# Learning to Transduce with Unbounded Memory

**Edward Grefenstette**
Google DeepMind
etg@google.com

**Karl Moritz Hermann**
Google DeepMind
kmh@google.com

**Mustafa Suleyman**
Google DeepMind
mustafasul@google.com

**Phil Blunsom**
Google DeepMind and Oxford University
pblunsom@google.com

## Abstract

Recently, strong results have been demonstrated by Deep Recurrent Neural Networks on natural language transduction problems. In this paper we explore the representational power of these models using synthetic grammars designed to exhibit phenomena similar to those found in real transduction problems such as machine translation. These experiments lead us to propose new memory-based recurrent networks that implement continuously differentiable analogues of traditional data structures such as Stacks, Queues, and DeQues. We show that these architectures exhibit superior generalisation performance to Deep RNNs and are often able to learn the underlying generating algorithms in our transduction experiments.

## 1 Introduction

Recurrent neural networks (RNNs) offer a compelling tool for processing natural language input in a straightforward sequential manner. Many natural language processing (NLP) tasks can be viewed as transduction problems, that is learning to convert one string into another. Machine translation is a prototypical example of transduction and recent results indicate that Deep RNNs have the ability to encode long source strings and produce coherent translations [1, 2]. While elegant, the application of RNNs to transduction tasks requires hidden layers large enough to store representations of the longest strings likely to be encountered, implying wastage on shorter strings and a strong dependency between the number of parameters in the model and its memory.

In this paper we use a number of linguistically-inspired synthetic transduction tasks to explore the ability of RNNs to learn long-range reorderings and substitutions. Further, inspired by prior work on neural network implementations of stack data structures [3], we propose and evaluate transduction models based on Neural Stacks, Queues, and DeQues (double ended queues). Stack algorithms are well-suited to processing the hierarchical structures observed in natural language and we hypothesise that their neural analogues will provide an effective and learnable transduction tool. Our models provide a middle ground between simple RNNs and the recently proposed Neural Turing Machine (NTM) [4] which implements a powerful random access memory with read and write operations. Neural Stacks, Queues, and DeQues also provide a logically unbounded memory while permitting efficient constant time push and pop operations.

Our results indicate that the models proposed in this work, and in particular the Neural DeQue, are able to consistently learn a range of challenging transductions. While Deep RNNs based on long short-term memory (LSTM) cells [1, 5] can learn some transductions when tested on inputs of the same length as seen in training, they fail to consistently generalise to longer strings. In contrast, our sequential memory-based algorithms are able to learn to reproduce the generating transduction algorithms, often generalising perfectly to inputs well beyond those encountered in training.

## 2 Related Work

String transduction is central to many applications in NLP, from name transliteration and spelling correction, to inflectional morphology and machine translation. The most common approach leverages symbolic finite state transducers [6, 7], with approaches based on context free representations also being popular [8]. RNNs offer an attractive alternative to symbolic transducers due to their simple algorithms and expressive representations [9]. However, as we show in this work, such models are limited in their ability to generalise beyond their training data and have a memory capacity that scales with the number of their trainable parameters.

Previous work has touched on the topic of rendering discrete data structures such as stacks continuous, especially within the context of modelling pushdown automata with neural networks [10, 11, 3]. We were inspired by the continuous pop and push operations of these architectures and the idea of an RNN controlling the data structure when developing our own models. The key difference is that our work adapts these operations to work within a recurrent continuous Stack/Queue/DeQue-like structure, the dynamics of which are fully decoupled from those of the RNN controlling it. In our models, the backwards dynamics are easily analysable in order to obtain the exact partial derivatives for use in error propagation, rather than having to approximate them as done in previous work.

In a parallel effort to ours, researchers are exploring the addition of memory to recurrent networks. The NTM and Memory Networks [4, 12, 13] provide powerful random access memory operations, whereas we focus on a more efficient and restricted class of models which we believe are sufficient for natural language transduction tasks. More closely related to our work, [14] have sought to develop a continuous stack controlled by an RNN. Note that this model—unlike the work proposed here—renders discrete push and pop operations continuous by "mixing" information across levels of the stack at each time step according to scalar push/pop action values. This means the model ends up compressing information in the stack, thereby limiting its use, as it effectively loses the unbounded memory nature of traditional symbolic models.

## 3 Models

In this section, we present an extensible memory enhancement to recurrent layers which can be set up to act as a continuous version of a classical Stack, Queue, or DeQue (double-ended queue). We begin by describing the operations and dynamics of a neural Stack, before showing how to modify it to act as a Queue, and extend it to act as a DeQue.

### 3.1 Neural Stack

Let a Neural Stack be a differentiable structure onto and from which continuous vectors are pushed and popped. Inspired by the neural pushdown automaton of [3], we render these traditionally discrete operations continuous by letting push and pop operations be real values in the interval $(0, 1)$. Intuitively, we can interpret these values as the *degree of certainty* with which some controller wishes to push a vector $v$ onto the stack, or pop the top of the stack.

$$V_t[i] = \begin{cases} V_{t-1}[i] & \text{if } 1 \leq i < t \\ \mathbf{v}_t & \text{if } i = t \end{cases} \quad (\text{Note that } V_t[i] = \mathbf{v}_i \text{ for all } i \leq t) \tag{1}$$

$$\mathbf{s}_t[i] = \begin{cases} max(0, \mathbf{s}_{t-1}[i] - max(0, u_t - \sum_{j=i+1}^{t-1} \mathbf{s}_{t-1}[j])) & \text{if } 1 \leq i < t \\ d_t & \text{if } i = t \end{cases} \tag{2}$$

$$\mathbf{r}_t = \sum_{i=1}^{t} (min(\mathbf{s}_t[i], max(0, 1 - \sum_{j=i+1}^{t} \mathbf{s}_t[j]))) \cdot V_t[i] \tag{3}$$

Formally, a Neural Stack, fully parametrised by an embedding size $m$, is described at some timestep $t$ by a $t \times m$ value matrix $V_t$ and a strength vector $\mathbf{s}_t \in \mathbb{R}^t$. These form the core of a recurrent layer which is acted upon by a controller by receiving, from the controller, a value $\mathbf{v}_t \in \mathbb{R}^m$, a pop signal $u_t \in (0, 1)$, and a push signal $d_t \in (0, 1)$. It outputs a read vector $\mathbf{r}_t \in \mathbb{R}^m$. The recurrence of this

layer comes from the fact that it will receive as previous state of the stack the pair $(V_{t-1}, \mathbf{s}_{t-1})$, and produce as next state the pair $(V_t, \mathbf{s}_t)$ following the dynamics described below. Here, $V_t[i]$ represents the $i$th row (an $m$-dimensional vector) of $V_t$ and $\mathbf{s}_t[i]$ represents the $i$th value of $\mathbf{s}_t$.

Equation 1 shows the update of the value component of the recurrent layer state represented as a matrix, the number of rows of which grows with time, maintaining a record of the values pushed to the stack at each timestep (whether or not they are still logically on the stack). Values are appended to the bottom of the matrix (top of the stack) and never changed.

Equation 2 shows the effect of the push and pop signal in updating the strength vector $\mathbf{s}_{t-1}$ to produce $\mathbf{s}_t$. First, the pop operation removes objects from the stack. We can think of the pop value $u_t$ as the initial deletion quantity for the operation. We traverse the strength vector $\mathbf{s}_{t-1}$ from the highest index to the lowest. If the next strength scalar is less than the remaining deletion quantity, it is subtracted from the remaining quantity and its value is set to $0$. If the remaining deletion quantity is less than the next strength scalar, the remaining deletion quantity is subtracted from that scalar and deletion stops. Next, the push value is set as the strength for the value added in the current timestep.

Equation 3 shows the dynamics of the read operation, which are similar to the pop operation. A fixed initial read quantity of $1$ is set at the top of a temporary copy of the strength vector $\mathbf{s}_t$ which is traversed from the highest index to the lowest. If the next strength scalar is smaller than the remaining read quantity, its value is preserved for this operation and subtracted from the remaining read quantity. If not, it is temporarily set to the remaining read quantity, and the strength scalars of all lower indices are temporarily set to $0$. The output $\mathbf{r}_t$ of the read operation is the weighted sum of the rows of $V_t$, scaled by the temporary scalar values created during the traversal. An example of the stack read calculations across three timesteps, after pushes and pops as described above, is illustrated in Figure 1a. The third step shows how setting the strength $\mathbf{s}_3[2]$ to $0$ for $V_3[2]$ logically removes $\mathbf{v}_2$ from the stack, and how it is ignored during the read.

This completes the description of the forward dynamics of a neural Stack, cast as a recurrent layer, as illustrated in Figure 1b. All operations described in this section are differentiable[1]. The equations describing the backwards dynamics are provided in Appendix A of the supplementary materials.

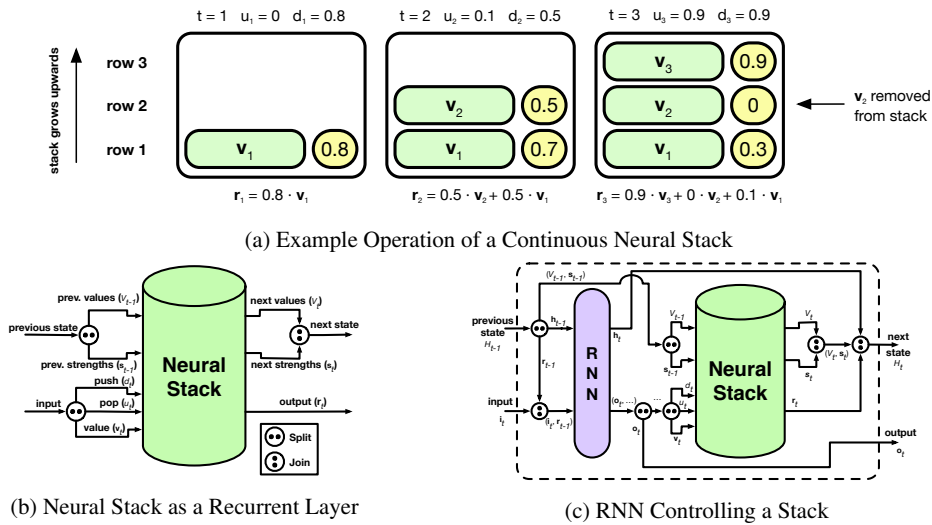

(a) Example Operation of a Continuous Neural Stack

(b) Neural Stack as a Recurrent Layer

(c) RNN Controlling a Stack

Figure 1: Illustrating a Neural Stack's Operations, Recurrent Structure, and Control

## 3.2   Neural Queue

A neural Queue operates the same way as a neural Stack, with the exception that the pop operation reads the lowest index of the strength vector $\mathbf{s}_t$, rather than the highest. This represents popping and

reading from the front of the Queue rather than the top of the stack. These operations are described in Equations 4–5.

$$\mathbf{s}_t[i] = \begin{cases} max(0, \mathbf{s}_{t-1}[i] - max(0, u_t - \sum_{j=1}^{i-1} \mathbf{s}_{t-1}[j])) & \text{if } 1 \le i < t \\ d_t & \text{if } i = t \end{cases} \tag{4}$$

$$\mathbf{r}_t = \sum_{i=1}^{t} (min(\mathbf{s}_t[i], max(0, 1 - \sum_{j=1}^{i-1} \mathbf{s}_t[j]))) \cdot V_t[i] \tag{5}$$

### 3.3 Neural DeQue

A neural DeQue operates likes a neural Stack, except it takes a push, pop, and value as input for both "ends" of the structure (which we call $top$ and $bot$), and outputs a read for both ends. We write $u_t^{top}$ and $u_t^{bot}$ instead of $u_t$, $\mathbf{v}_t^{top}$ and $\mathbf{v}_t^{bot}$ instead of $\mathbf{v}_t$, and so on. The state, $V_t$ and $\mathbf{s}_t$ are now a $2t \times m$-dimensional matrix and a $2t$-dimensional vector, respectively. At each timestep, a pop from the top is followed by a pop from the bottom of the DeQue, followed by the pushes and reads. The dynamics of a DeQue, which unlike a neural Stack or Queue "grows" in two directions, are described in Equations 6–11, below. Equations 7–9 decompose the strength vector update into three steps purely for notational clarity.

$$V_t[i] = \begin{cases} \mathbf{v}_t^{bot} & \text{if } i = 1 \\ \mathbf{v}_t^{top} & \text{if } i = 2t \\ V_{t-1}[i-1] & \text{if } 1 < i < 2t \end{cases} \tag{6}$$

$$\mathbf{s}_t^{top}[i] = max(0, \mathbf{s}_{t-1}[i] - max(0, u_t^{top} - \sum_{j=i+1}^{2(t-1)-1} \mathbf{s}_{t-1}[j])) \quad \text{if } 1 \le i < 2(t-1) \tag{7}$$

$$\mathbf{s}_t^{both}[i] = max(0, \mathbf{s}_t^{top}[i] - max(0, u_t^{bot} - \sum_{j=1}^{i-1} \mathbf{s}_t^{top}[j])) \quad \text{if } 1 \le i < 2(t-1) \tag{8}$$

$$\mathbf{s}_t[i] = \begin{cases} \mathbf{s}_t^{both}[i-1] & \text{if } 1 < i < 2t \\ d_t^{bot} & \text{if } i = 1 \\ d_t^{top} & \text{if } i = 2t \end{cases} \tag{9}$$

$$\mathbf{r}_t^{top} = \sum_{i=1}^{2t} (min(\mathbf{s}_t[i], max(0, 1 - \sum_{j=i+1}^{2t} \mathbf{s}_t[j]))) \cdot V_t[i] \tag{10}$$

$$\mathbf{r}_t^{bot} = \sum_{i=1}^{2t} (min(\mathbf{s}_t[i], max(0, 1 - \sum_{j=1}^{i-1} \mathbf{s}_t[j]))) \cdot V_t[i] \tag{11}$$

To summarise, a neural DeQue acts like two neural Stacks operated on in tandem, except that the pushes and pops from one end may eventually affect pops and reads on the other, and vice versa.

### 3.4 Interaction with a Controller

While the three memory modules described can be seen as recurrent layers, with the operations being used to produce the next state and output from the input and previous state being fully differentiable, they contain no tunable parameters to optimise during training. As such, they need to be attached to a controller in order to be used for any practical purposes. In exchange, they offer an extensible memory, the logical size of which is unbounded and decoupled from both the nature and parameters of the controller, and from the size of the problem they are applied to. Here, we describe how any RNN controller may be enhanced by a neural Stack, Queue or DeQue.

We begin by giving the case where the memory is a neural Stack, as illustrated in Figure 1c. Here we wish to replicate the overall 'interface' of a recurrent layer—as seen from outside the dotted

lines—which takes the previous recurrent state $H_{t-1}$ and an input vector $\mathbf{i}_t$, and transforms them to return the next recurrent state $H_t$ and an output vector $\mathbf{o}_t$. In our setup, the previous state $H_{t-1}$ of the recurrent layer will be the tuple $(\mathbf{h}_{t-1}, \mathbf{r}_{t-1}, (V_{t-1}, \mathbf{s}_{t-1}))$, where $\mathbf{h}_{t-1}$ is the previous state of the RNN, $\mathbf{r}_{t-1}$ is the previous stack read, and $(V_{t-1}, \mathbf{s}_{t-1})$ is the previous state of the stack as described above. With the exception of $\mathbf{h}_0$, which is initialised randomly and optimised during training, all other initial states, $\mathbf{r}_0$ and $(V_0, \mathbf{s}_0)$, are set to 0-valued vectors/matrices and not updated during training.

The overall input $\mathbf{i}_t$ is concatenated with previous read $\mathbf{r}_{t-1}$ and passed to the RNN controller as input along with the previous controller state $\mathbf{h}_{t-1}$. The controller outputs its next state $\mathbf{h}_t$ and a controller output $\mathbf{o}'_t$, from which we obtain the push and pop scalars $d_t$ and $u_t$ and the value vector $\mathbf{v_t}$, which are passed to the stack, as well as the network output $\mathbf{o}_t$:

$$d_t = sigmoid(W_d \mathbf{o}'_t + b_d) \qquad\qquad u_t = sigmoid(W_u \mathbf{o}'_t + b_u)$$
$$\mathbf{v}_t = tanh(W_v \mathbf{o}'_t + \mathbf{b}_v) \qquad\qquad \mathbf{o}_t = tanh(W_o \mathbf{o}'_t + \mathbf{b}_o)$$

where $W_d$ and $W_u$ are vector-to-scalar projection matrices, and $b_d$ and $b_u$ are their scalar biases; $W_v$ and $W_o$ are vector-to-vector projections, and $\mathbf{b}_d$ and $\mathbf{b}_u$ are their vector biases, all randomly intialised and then tuned during training. Along with the previous stack state $(V_{t-1}, \mathbf{s}_{t-1})$, the stack operations $d_t$ and $u_t$ and the value $\mathbf{v_t}$ are passed to the neural stack to obtain the next read $\mathbf{r}_t$ and next stack state $(V_t, \mathbf{s}_t)$, which are packed into a tuple with the controller state $\mathbf{h_t}$ to form the next state $H_t$ of the overall recurrent layer. The output vector $\mathbf{o}_t$ serves as the overall output of the recurrent layer. The structure described here can be adapted to control a neural Queue instead of a stack by substituting one memory module for the other.

The only additional trainable parameters in either configuration, relative to a non-enhanced RNN, are the projections for the input concatenated with the previous read into the RNN controller, and the projections from the controller output into the various Stack/Queue inputs, described above. In the case of a DeQue, both the top read $\mathbf{r}^{top}$ and bottom read $\mathbf{r}^{bot}$ must be preserved in the overall state. They are both concatenated with the input to form the input to the RNN controller. The output of the controller must have additional projections to output push/pop operations and values for the bottom of the DeQue. This roughly doubles the number of additional tunable parameters "wrapping" the RNN controller, compared to the Stack/Queue case.

## 4  Experiments

In every experiment, integer-encoded source and target sequence pairs are presented to the candidate model as a batch of single joint sequences. The joint sequence starts with a start-of-sequence (SOS) symbol, and ends with an end-of-sequence (EOS) symbol, with a separator symbol separating the source and target sequences. Integer-encoded symbols are converted to $64$-dimensional embeddings via an embedding matrix, which is randomly initialised and tuned during training. Separate word-to-index mappings are used for source and target vocabularies. Separate embedding matrices are used to encode input and output (predicted) embeddings.

### 4.1  Synthetic Transduction Tasks

The aim of each of the following tasks is to read an input sequence, and generate as target sequence a transformed version of the source sequence, followed by an EOS symbol. Source sequences are randomly generated from a vocabulary of 128 meaningless symbols. The length of each training source sequence is uniformly sampled from $unif\{8, 64\}$, and each symbol in the sequence is drawn with replacement from a uniform distribution over the source vocabulary (ignoring SOS, and separator).

A deterministic task-specific transformation, described for each task below, is applied to the source sequence to yield the target sequence. As the training sequences are entirely determined by the source sequence, there are close to $10^{135}$ training sequences for each task, and training examples are sampled from this space due to the random generation of source sequences. The following steps are followed before each training and test sequence are presented to the models, the SOS symbol ($\langle s \rangle$) is prepended to the source sequence, which is concatenated with a separator symbol ($|||$) and the target sequences, to which the EOS symbol ($\langle /s \rangle$) is appended.

**Sequence Copying**    The source sequence is copied to form the target sequence. Sequences have the form:

$$\langle s \rangle a_1 \ldots a_k || a_1 \ldots a_k \langle /s \rangle$$

**Sequence Reversal**    The source sequence is deterministically reversed to produce the target sequence. Sequences have the form:

$$\langle s \rangle a_1 a_2 \ldots a_k || a_k \ldots a_2 a_1 \langle /s \rangle$$

**Bigram flipping**    The source side is restricted to even-length sequences. The target is produced by swapping, for all odd source sequence indices $i \in [1, |seq|] \wedge odd(i)$, the $i$th symbol with the $(i+1)$th symbol. Sequences have the form:

$$\langle s \rangle a_1 a_2 a_3 a_4 \ldots a_{k-1} a_k || a_2 a_1 a_4 a_3 \ldots a_k a_{k-1} \langle /s \rangle$$

### 4.2    ITG Transduction Tasks

The following tasks examine how well models can approach sequence transduction problems where the source and target sequence are jointly generated by Inversion Transduction Grammars (ITG) [8], a subclass of Synchronous Context-Free Grammars [16] often used in machine translation [17]. We present two simple ITG-based datasets with interesting linguistic properties and their underlying grammars. We show these grammars in Table 1, in Appendix C of the supplementary materials. For each synchronised non-terminal, an expansion is chosen according to the probability distribution specified by the rule probability $p$ at the beginning of each rule. For each grammar, 'A' is always the root of the ITG tree.

We tuned the generative probabilities for recursive rules by hand so that the grammars generate left and right sequences of lengths 8 to 128 with relatively uniform distribution. We generate training data by rejecting samples that are outside of the range $[8, 64]$, and testing data by rejecting samples outside of the range $[65, 128]$. For terminal symbol-generating rules, we balance the classes so that for $k$ terminal-generating symbols in the grammar, each terminal-generating non-terminal 'X' generates a vocabulary of approximately $128/k$, and each each vocabulary word under that class is equiprobable. These design choices were made to maximise the similarity between the experimental settings of the ITG tasks described here and the synthetic tasks described above.

**Subj–Verb–Obj to Subj–Obj–Verb**    A persistent challenge in machine translation is to learn to faithfully reproduce high-level syntactic divergences between languages. For instance, when translating an English sentence with a non-finite verb into German, a transducer must locate and move the verb over the object to the final position. We simulate this phenomena with a synchronous grammar which generates strings exhibiting verb movements. To add an extra challenge, we also simulate simple relative clause embeddings to test the models' ability to transduce in the presence of unbounded recursive structures.

A sample output of the grammar is presented here, with spaces between words being included for stylistic purposes, and where s, o, and v indicate subject, object, and verb terminals respectively, i and o mark input and output, and rp indicates a relative pronoun:

  si1 vi28 oi5 oi7 si15 rpi si19 vi16 oi10 oi24 ||| so1 oo5 oo7 so15 rpo so19 vo16 oo10 oo24 vo28

**Genderless to gendered grammar**    We design a small grammar to simulate translations from a language with gender-free articles to one with gender-specific definite and indefinite articles. A real world example of such a translation would be from English (*the, a*) to German (*der/die/das, ein/eine/ein*).

The grammar simulates sentences in $(NP/(V/NP))$ or $(NP/V)$ form, where every noun phrase can become an infinite sequence of nouns joined by a conjunction. Each noun in the source language has a neutral definite or indefinite article. The matching word in the target language then needs to be preceded by its appropriate article. A sample output of the grammar is presented here, with spaces between words being included for stylistic purposes:

  we11 the en19 and the em17 ||| wg11 das gn19 und der gm17

### 4.3 Evaluation

For each task, test data is generated through the same procedure as training data, with the key difference that the length of the source sequence is sampled from $unif\{65, 128\}$. As a result of this change, we not only are assured that the models cannot observe any test sequences during training, but are also measuring how well the sequence transduction capabilities of the evaluated models generalise beyond the sequence lengths observed during training. To control for generalisation ability, we also report accuracy scores on sequences separately sampled from the training set, which given the size of the sample space are unlikely to have ever been observed during actual model training.

For each round of testing, we sample 1000 sequences from the appropriate test set. For each sequence, the model reads in the source sequence and separator symbol, and begins generating the next symbol by taking the maximally likely symbol from the softmax distribution over target symbols produced by the model at each step. Based on this process, we give each model a *coarse accuracy* score, corresponding to the proportion of test sequences correctly predicted from beginning until end (EOS symbol) without error, as well as a *fine accuracy* score, corresponding to the average proportion of each sequence correctly generated before the first error. Formally, we have:

$$coarse = \frac{\#correct}{\#seqs} \qquad fine = \frac{1}{\#seqs} \sum_{i=1}^{\#seqs} \frac{\#correct_i}{|target_i|}$$

where $\#correct$ and $\#seqs$ are the number of correctly predicted sequences (end-to-end) and the total number of sequences in the test batch (1000 in this experiment), respectively; $\#correct_i$ is the number of correctly predicted symbols before the first error in the $i$th sequence of the test batch, and $|target_i|$ is the length of the target segment that sequence (including EOS symbol).

### 4.4 Models Compared and Experimental Setup

For each task, we use as benchmarks the Deep LSTMs described in [1], with 1, 2, 4, and 8 layers. Against these benchmarks, we evaluate neural Stack-, Queue-, and DeQue-enhanced LSTMs. When running experiments, we trained and tested a version of each model where all LSTMs in each model have a hidden layer size of 256, and one for a hidden layer size of 512. The Stack/Queue/DeQue embedding size was arbitrarily set to 256, half the maximum hidden size. The number of parameters for each model are reported for each architecture in Table 2 of the appendix. Concretely, the neural Stack-, Queue-, and DeQue-enhanced LSTMs have the same number of trainable parameters as a two-layer Deep LSTM. These all come from the extra connections to and from the memory module, which itself has no trainable parameters, regardless of its logical size.

Models are trained with minibatch RMSProp [18], with a batch size of 10. We grid-searched learning rates across the set $\{5 \times 10^{-3}, 1 \times 10^{-3}, 5 \times 10^{-4}, 1 \times 10^{-4}, 5 \times 10^{-5}\}$. We used gradient clipping [19], clipping all gradients above 1. Average training perplexity was calculated every 100 batches. Training and test set accuracies were recorded every 1000 batches.

## 5 Results and Discussion

Because of the impossibility of overfitting the datasets, we let the models train an unbounded number of steps, and report results at convergence. We present in Figure 2a the coarse- and fine-grained accuracies, for each task, of the best model of each architecture described in this paper alongside the best performing Deep LSTM benchmark. The best models were automatically selected based on average training perplexity. The LSTM benchmarks performed similarly across the range of random initialisations, so the effect of this procedure is primarily to try and select the better performing Stack/Queue/DeQue-enhanced LSTM. In most cases, this procedure does not yield the actual best-performing model, and in practice a more sophisticated procedure such as ensembling [20] should produce better results.

For all experiments, the Neural Stack or Queue outperforms the Deep LSTM benchmarks, often by a significant margin. For most experiments, if a Neural Stack- or Queue-enhanced LSTM learns to partially or consistently solve the problem, then so does the Neural DeQue. For experiments where the enhanced LSTMs solve the problem completely (consistent accuracy of 1) in training, the accuracy persists in longer sequences in the test set, whereas benchmark accuracies drop for

| Experiment | Model | Training | | Testing | |
|---|---|---|---|---|---|
| | | Coarse | Fine | Coarse | Fine |
| Sequence Copying | 4-layer LSTM | 0.98 | 0.98 | 0.01 | 0.50 |
| | Stack-LSTM | 0.89 | 0.94 | 0.00 | 0.22 |
| | **Queue-LSTM** | **1.00** | **1.00** | **1.00** | **1.00** |
| | **DeQue-LSTM** | **1.00** | **1.00** | **1.00** | **1.00** |
| Sequence Reversal | 8-layer LSTM | 0.95 | 0.98 | 0.04 | 0.13 |
| | **Stack-LSTM** | **1.00** | **1.00** | **1.00** | **1.00** |
| | Queue-LSTM | 0.44 | 0.61 | 0.00 | 0.01 |
| | **DeQue-LSTM** | **1.00** | **1.00** | **1.00** | **1.00** |
| Bigram Flipping | 2-layer LSTM | 0.54 | 0.93 | 0.02 | 0.52 |
| | Stack-LSTM | 0.44 | 0.90 | 0.00 | 0.48 |
| | **Queue-LSTM** | **0.55** | **0.94** | **0.55** | **0.98** |
| | **DeQue-LSTM** | **0.55** | **0.94** | **0.53** | **0.98** |
| SVO to SOV | 8-layer LSTM | 0.98 | 0.99 | 0.98 | 0.99 |
| | **Stack-LSTM** | **1.00** | **1.00** | **1.00** | **1.00** |
| | **Queue-LSTM** | **1.00** | **1.00** | **1.00** | **1.00** |
| | **DeQue-LSTM** | **1.00** | **1.00** | **1.00** | **1.00** |
| Gender Conjugation | 8-layer LSTM | 0.98 | 0.99 | 0.99 | 0.99 |
| | Stack-LSTM | 0.93 | 0.97 | 0.93 | 0.97 |
| | **Queue-LSTM** | **1.00** | **1.00** | **1.00** | **1.00** |
| | **DeQue-LSTM** | **1.00** | **1.00** | **1.00** | **1.00** |

(a) Comparing Enhanced LSTMs to Best Benchmarks

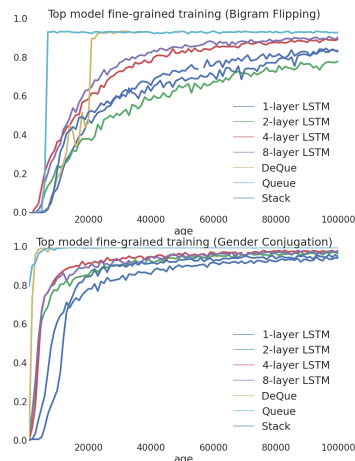

(b) Comparison of Model Convergence during Training

Figure 2: Results on the transduction tasks and convergence properties

all experiments except the SVO to SOV and Gender Conjugation ITG transduction tasks. Across all tasks which the enhanced LSTMs solve, the convergence on the top accuracy happens orders of magnitude earlier for enhanced LSTMs than for benchmark LSTMs, as exemplified in Figure 2b.

The results for the sequence inversion and copying tasks serve as unit tests for our models, as the controller mainly needs to learn to push the appropriate number of times and then pop continuously. Nonetheless, the failure of Deep LSTMs to learn such a regular pattern and generalise is itself indicative of the limitations of the benchmarks presented here, and of the relative expressive power of our models. Their ability to generalise perfectly to sequences up to twice as long as those attested during training is also notable, and also attested in the other experiments. Finally, this pair of experiments illustrates how while the neural Queue solves copying and the Stack solves reversal, a simple LSTM controller can learn to operate a DeQue as either structure, and solve both tasks.

The results of the Bigram Flipping task for all models are consistent with the failure to consistently correctly generate the last two symbols of the sequence. We hypothesise that both Deep LSTMs and our models economically learn to pairwise flip the sequence tokens, and attempt to do so half the time when reaching the EOS token. For the two ITG tasks, the success of Deep LSTM benchmarks relative to their performance in other tasks can be explained by their ability to exploit short local dependencies dominating the longer dependencies in these particular grammars.

Overall, the rapid convergence, where possible, on a general solution to a transduction problem in a manner which propagates to longer sequences without loss of accuracy is indicative that an unbounded memory-enhanced controller can learn to solve these problems procedurally, rather than memorising the underlying distribution of the data.

# 6   Conclusions

The experiments performed in this paper demonstrate that single-layer LSTMs enhanced by an unbounded differentiable memory capable of acting, in the limit, like a classical Stack, Queue, or DeQue, are capable of solving sequence-to-sequence transduction tasks for which Deep LSTMs falter. Even in tasks for which benchmarks obtain high accuracies, the memory-enhanced LSTMs converge earlier, and to higher accuracies, while requiring considerably fewer parameters than all but the simplest of Deep LSTMs. We therefore believe these constitute a crucial addition to our neural network toolbox, and that more complex linguistic transduction tasks such as machine translation or parsing will be rendered more tractable by their inclusion.

## Footnotes

[1]The $max(x, y)$ and $min(x, y)$ functions are technically not differentiable for $x = y$. Following the work on rectified linear units [15], we arbitrarily take the partial differentiation of the left argument in these cases.

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
