[Supplementary Material]

## A Analysis of the Backwards Dynamics of a Neural Stack

We describe here the backwards dynamics of the neural stack by examining the relevant partial derivatives of of the outputs with regard to the inputs, as defined in Equations 1–3. We use $\delta_{ij}$ to indicate the Kronecker delta (1 if $i = j$, 0 otherwise). The equations below hold for any valid row numbers $i$ and $n$.

$$\frac{\partial V_t[i]}{\partial V_{t-1}[n]} = \delta_{in} \qquad (12) \qquad \frac{\partial V_t[i]}{\partial \mathbf{v}_t} = \delta_{it} \qquad (13) \qquad \frac{\partial \mathbf{s}_t[i]}{\partial d_t} = \delta_{it} \qquad (14)$$

$$\frac{\partial \mathbf{r}_t}{\partial V_t[n]} = min(\mathbf{s}_t[n], max(0, 1 - \sum_{j=n+1}^{t} \mathbf{s}_t[j])) \quad \text{and} \quad \frac{\partial \mathbf{r}_t}{\partial \mathbf{v}_t} = \frac{\partial \mathbf{r}_t}{\partial V_t[t]} = d_t \qquad (15)$$

$$\frac{\partial \mathbf{s}_t[i]}{\partial \mathbf{s}_{t-1}[n]} = \begin{cases} 1 & \text{if } i < n < t \text{ and } \mathbf{s}_t[i] > 0 \text{ and } u_t - \sum_{j=i+1}^{t-1} \mathbf{s}_{t-1}[j] > 0 \\ \delta_{in} & \text{if } i < t \text{ and } \mathbf{s}_t[i] > 0 \text{ and } u_t - \sum_{j=i+1}^{t-1} \mathbf{s}_{t-1}[j] \leq 0 \\ 0 & \text{otherwise} \end{cases} \qquad (16)$$

$$\frac{\partial \mathbf{s}_t[i]}{\partial u_t} = \begin{cases} \text{-}1 & \text{if } i < t \text{ and } \mathbf{s}_t[i] > 0 \text{ and } u_t - \sum_{j=i+1}^{t-1} \mathbf{s}_{t-1}[j] > 0 \\ 0 & \text{otherwise} \end{cases} \qquad (17)$$

$$\frac{\partial \mathbf{r}_t}{\partial \mathbf{s}_t[n]} = \sum_{i=1}^{t} h(i,n) \cdot V_t[i]$$

$$\text{where} \quad h(i,n) = \begin{cases} \delta_{in} & \text{if } \mathbf{s}_t[i] \leq max(0, 1 - \sum_{j=i+1}^{t} \mathbf{s}_t[j]) \\ -1 & \text{if } i < n \text{ and } \mathbf{s}_t[i] > max(0, 1 - \sum_{j=i+1}^{t} \mathbf{s}_t[j]) \\ & \text{and } \sum_{j=i+1}^{t} \mathbf{s}_t[j] \leq 1 \\ 0 & \text{otherwise} \end{cases} \qquad (18)$$

All partial derivatives other than those obtained by the chain rule for derivatives can be assumed to be 0. The backwards dynamics for neural Queues and DeQues can be similarly derived from Equations 4–11.

## B A Note on Controller Initialisation

During initial experiments with the continuous stack presented in this paper, we noted that the stack's ability to learn the solution to the transduction tasks detailed here varied greatly based on the random initialisation of the controller. This initially required us to restart training with different random seeds to obtain behaviour consistent with the learning of an algorithmic solution (i.e. rapid drop in validation perplexity after a short number of iterations).

Analysis of the backwards dynamics presented in Section A demonstrates that error on push and pop decisions is a function of read error "carried" back through time by the vectors on the stack (*cf.* Equation 14 and Equations 17–18), which is accumulated as the vectors placed onto the stack by a push, or retained after a pop, are read at further timesteps. Crucially, this means that if the controller operating the stack is initially biased in favour of popping over pushing (i.e. $u_{t+1} > d_t$ for most or all timesteps $t$), vectors are likely to be removed from the stack the timestep after they were pushed, resulting in the continuous stack being used as an extra recurrent hidden layer, rather than as something behaving like a classical stack.

The consequence of this is that gradient for the decision to push at time $t$ only comes via the hidden state of the controller at time $t + 1$, so for problems where the vector would ideally have been

preserved on the stack until some later time, signal encouraging the controller to push with higher certainty is unlikely to be propagated back if the RNN controller suffers from vanishing gradient issues. Likewise, the gradient for the decision to pop is 0 (as each pop empties the stack). We conclude that under-using the memory in such a way makes its proper manipulation hard to learn by the controller.

Conversely, over-using the stack (even incorrectly) means that gradient obtained with regard to the (mis)use is properly communicated, as the pop gradient will not be zero (Equation 17) for all $t$. Additionally, the (non-vanishing) gradient propagated through the stack state (Equation 12) will allow the decision to push at some timestep to be rewarded or penalised based on reads at some much later time. These remarks also apply to the continuous queue and double-ended queue.

Since in our setting the decision to push and pop is produced by taking a biased linear transform of an RNN hidden state followed by a component-wise sigmoid operation, we hypothesised, based on the above analysis, that initialising the bias for popping to a negative number would solve the variance issue described above. We tested this on short sequences of the copy task, and found that a small bias of $-1$ produced the desired algorithmic behaviour of the stack-enhanced controller across all seeds tested. Setting this initialisation policy for the controller across all experiments allowed us to reproduce the results produced in the paper without need for repeated initialisation. We recommend that other controller implementations provide similar trainable biases for the decision to pop, and initialise them following this policy (and likewise for controllers controlling other continuous data structures presented in this paper).

## C   Inversion Transduction Grammars used in Experiments

We present here, in Table 1, the inverse transduction grammars described in Section 4.2. Sets of terminal-generating rules are indicated by the form '$X_i \to \ldots$', where $i \in [1, k]$ and $p(X_i) \approx (100/k)^{-1}$ for $k$ terminal generating non-terminal symbols (classes of terminals), so that the generated vocabulary is balanced across classes and of a size similar to other experiments.

| $p$ | ITG Rules |
|---|---|
| 1 | A → S1 VT2 O3 \| S1 O3 VT2 |
| 1/5 | S → S1 S2 \| S1 S2 |
| 1/5 | S → S1 rpi S2 VT3 \| S1 rpo S2 VT3 |
| 3/5 | S → ST1 \| ST1 |
| 1/5 | O → O1 O2 \| O1 O2 |
| 1/5 | O → S1 rpi S2 VT3 \| S1 rpo S2 VT3 |
| 3/5 | O → OT1 \| OT1 |
| 1/33 | $ST_i$ → $si_i$ \| $so_i$ |
| 1/33 | $OT_i$ → $oi_i$ \| $oo_i$ |
| 1/33 | $VT_i$ → $vi_i$ \| $vo_i$ |

(a) SVO-SOV Grammar

| $p$ | ITG Rules |
|---|---|
| 1 | A → B1 \| B1 |
| 1/4 | B → B1 or B2 \| B1 oder B2 |
| 1/4 | B → S1 and S2 \| S1 und S2 |
| 1/2 | B → B1 V1 \| B1 V1 |
| 3/4 | V → W1 B2 \| W1 B2 |
| 1/4 | V → W1 \| W1 |
| 1/6 | S → the M1 \| der M1 |
| 1/6 | S → the F1 \| die F1 |
| 1/6 | S → the N1 \| das N1 |
| 1/6 | S → a M1 \| ein M1 |
| 1/6 | S → a F1 \| eine F1 |
| 1/6 | S → a N1 \| ein N1 |
| 1/25 | $W_i$ → $we_i$ \| $wg_i$ |
| 1/25 | $M_i$ → $me_i$ \| $mg_i$ |
| 1/25 | $F_i$ → $fe_i$ \| $fg_i$ |
| 1/25 | $N_i$ → $ne_i$ \| $ng_i$ |

(b) English-German Conjugation Grammar

Table 1: Inversion Transduction Grammars used in ITG Tasks

## D   Model Sizes

We show, in Table 2, the number of parameters per model, for all models used in the experiments of the paper.

|  | Hidden layer size | |
|---|---|---|
| **Model** | **256** | **512** |
| 1-layer LSTM | $3.3 \times 10^5$ | $1.2 \times 10^6$ |
| 2-layer LSTM | $9.1 \times 10^5$ | $3.4 \times 10^6$ |
| 4-layer LSTM | $2.1 \times 10^6$ | $7.8 \times 10^6$ |
| 8-layer LSTM | $4.5 \times 10^6$ | $1.7 \times 10^7$ |
| Stack-LSTM | $6.7 \times 10^5$ | $1.9 \times 10^6$ |
| Queue-LSTM | $6.7 \times 10^5$ | $1.9 \times 10^6$ |
| DeQue-LSTM | $1.0 \times 10^6$ | $2.5 \times 10^6$ |

Table 2: Number of trainable parameters per model

# E  Full Results

We show in Table 3 the full results for each task of the best performing models. The procedure for selecting the best performing model is described in Section 5.

| | | Training | | Testing | |
|---|---|---|---|---|---|
| **Experiment** | **Model** | **Coarse** | **Fine** | **Coarse** | **Fine** |
| | 1-layer LSTM | 0.62 | 0.87 | 0.00 | 0.38 |
| | 2-layer LSTM | 0.80 | 0.95 | 0.00 | 0.47 |
| | 4-layer LSTM | 0.98 | 0.98 | 0.01 | 0.50 |
| **Sequence Copying** | 8-layer LSTM | 0.57 | 0.83 | 0.00 | 0.31 |
| | Stack-LSTM | 0.89 | 0.94 | 0.00 | 0.22 |
| | **Queue-LSTM** | **1.00** | **1.00** | **1.00** | **1.00** |
| | **DeQue-LSTM** | **1.00** | **1.00** | **1.00** | **1.00** |
| | 1-layer LSTM | 0.78 | 0.87 | 0.01 | 0.09 |
| | 2-layer LSTM | 0.91 | 0.94 | 0.02 | 0.06 |
| | 4-layer LSTM | 0.93 | 0.96 | 0.03 | 0.15 |
| **Sequence Reversal** | 8-layer LSTM | 0.95 | 0.98 | 0.04 | 0.13 |
| | **Stack-LSTM** | **1.00** | **1.00** | **1.00** | **1.00** |
| | Queue-LSTM | 0.44 | 0.61 | 0.00 | 0.07 |
| | **DeQue-LSTM** | **1.00** | **1.00** | **1.00** | **1.00** |
| | 1-layer LSTM | 0.53 | 0.93 | 0.01 | 0.53 |
| | 2-layer LSTM | 0.54 | 0.93 | 0.02 | 0.52 |
| | 4-layer LSTM | 0.52 | 0.93 | 0.01 | 0.56 |
| **Bigram Flipping** | 8-layer LSTM | 0.52 | 0.93 | 0.01 | 0.53 |
| | Stack-LSTM | 0.44 | 0.90 | 0.00 | 0.48 |
| | **Queue-LSTM** | **0.55** | **0.94** | **0.55** | **0.98** |
| | **DeQue-LSTM** | **0.55** | **0.94** | **0.53** | **0.98** |
| | 1-layer LSTM | 0.96 | 0.98 | 0.96 | 0.99 |
| | 2-layer LSTM | 0.97 | 0.99 | 0.96 | 0.99 |
| | 4-layer LSTM | 0.97 | 0.99 | 0.97 | 0.99 |
| **SVO to SOV** | 8-layer LSTM | 0.98 | 0.99 | 0.98 | 0.99 |
| | **Stack-LSTM** | **1.00** | **1.00** | **1.00** | **1.00** |
| | **Queue-LSTM** | **1.00** | **1.00** | **1.00** | **1.00** |
| | **DeQue-LSTM** | **1.00** | **1.00** | **1.00** | **1.00** |
| | 1-layer LSTM | 0.97 | 0.99 | 0.97 | 0.99 |
| | 2-layer LSTM | 0.98 | 0.99 | 0.98 | 0.99 |
| | 4-layer LSTM | 0.98 | 0.99 | 0.98 | 0.99 |
| **Gender Conjugation** | 8-layer LSTM | 0.98 | 0.99 | 0.99 | 0.99 |
| | Stack-LSTM | 0.93 | 0.97 | 0.93 | 0.97 |
| | **Queue-LSTM** | **1.00** | **1.00** | **1.00** | **1.00** |
| | **DeQue-LSTM** | **1.00** | **1.00** | **1.00** | **1.00** |

Table 3: Summary of Results for Transduction Tasks