[Reviews · NeurIPS 2015]

Submitted by Assigned_Reviewer_1

This paper demonstrates that deep recurrent neural networks are not able to model non-finite-state tasks, and proposes a memory-augmented RNN architecture with a novel way to make the memory operations continuous.

The demonstration is done using artificial grammars, which has the advantage that the properties in question are clear, but the disadvantage that it does not address the possibility that real natural language transduction problems don't actually have those properties.

It is nice to see an explicit demonstration of the fact that RNNs, even large deep ones, have finite memories.

This eliminates one of the possible explanations for why RNNs have shown such surprising abilities to deal with language transduction tasks, such as machine translation.

The proposed augmentations of RNN models with unbounded memory structures are similar to previous proposals of this kind, but novel in the way in which they are made continuous/differentiable.

This proposal for how to incorporate discrete data structures (stacks, queues, dequeues) into a NN shares with previous similar proposals a weakness in the motivation for the architecture.

It isn't clear why the descete data structure operations need to be made differentiable, and if so why in this way.

Why not just make the choice of operation nondeterministic and use sampling?

That would be the obvious thing to do, but no such baseline is evaluated.

This paper tries to motivate their proposal as an approximation to such an architecture:

"Intuitively, we can interpret these values as the degree of certainty with which some controller wishes to push a vector v onto the stack, or pop the top of the stack." But in fact the proposal does not correspond to such an approximation in any precise way.

The other main criticism is that no natural experiments are run, only artificial ones.

This leave open anaswered questions about whether the properties demonstrated in the artificial experiments actually apply in the target NLP applications.

The artificial experiments do constitute a contribution, but a less substantial one than I would expect to see in a NIPS paper.
Summary: This paper is an interesting investigation of the modelling power of deep recurrent neural networks, versus augmenting neural networks with memory, with the evaluation using artificial grammars.

Results are fairly predictable: deep RNNs are not able to encode an unbounded amount of information in a parametric vector space, but adding memory fixes that.

Submitted by Assigned_Reviewer_2

The paper describes augmentation of classical recursive networks by new memory models based on differentiable stack, queue and deque data structures to improve the ability of the network to effectively handle transduction tasks.

The paper is clearly written and easy to follow. The idea of differentiable stack is not completely new and is motivated on the former work of "push-down automata" by Sun et. al 1998 as correctly referenced in the paper. The queue and deque are however new and possibly useful for real transduction problems.

The main disadvantage of the paper is the evaluation which was done only on toy-problems and it is therefore hard to judge it's effectiveness on real problems. The paper would be much stronger if it for example showed empirical improvement on some of the natural language tasks such as mentioned machine translation.
Summary: The paper describes augmentation of classical recursive networks by new memory models based on differentiable stack, queue and deque data structures.

It's an interesting model but there are no experiments on real datasets or tasks.

Submitted by Assigned_Reviewer_3

TL;DR Using RNNs as sequence transducers is inefficient as they require sufficient capacity to store the entirety of the input sequence. This paper follows a line of work in which neural networks are augmented with external memory sources in such a way that they remain differentiable, and therefore trainable. The Neural Turing Machine (NTM) is one recent example. This paper proposes Neural Stacks, Queues, and DeQues, which allow unbounded memory but remain efficient for transduction problems.

Overall, I think this is quite a solid paper. My main gripe is that I would have liked to see discussion and experimental comparisons to RNNs augmented with attention / alignment models, such as in

Neural machine translation by jointly learning to align and translate D Bahdanau, K Cho, Y Bengio - arXiv preprint arXiv:1409.0473, 2014 - arxiv.org

Attention models appear to provide a solution to the problem of needing to encode the entire input sentence, and have already demonstrated gains on benchmark translation tasks. That said, I suspect the proposed neural data structures are able to generalize better, and may require less training data. On the other hand, it seems that the neural data structures sometimes impose a hurtful bias (e.g. some of synthetic tasks), so perhaps more care is needed when deciding to use them compared to a vanilla RNN model.

Regarding the experiments, it would have been nice to see results on standard benchmark tasks, even if they are smaller scale than MT (e.g. spelling correction, morphology). This fairly minor issue aside, I was also surprised not to see any plots comparing performance vs. a baseline RNN as sequence length varies, as this was another cited motivation for the paper. Presumably these plots would show that performance of the vanilla RNNs decreases as sequence length increases, while the augmented RNNs do not exhibit this issue.
Summary: This paper describes three models that serve as differentiable "neural" analogues of stacks, queues, and deques. The contributions are well-motivated, the presentation clear, and the experimental evaluation (though only on synthetic data) is fairly convincing.

Submitted by Assigned_Reviewer_4

This paper presents neural stack, queue, and neural deque data structures with the aim of drawing the 'attention' of the network to the relevant memory/information in the past. More specifically, the state of the data structure (its memory content and the strength signal) is updated by control signals from a 'controller'. The controller is the core part of the model which 'reads' the relevant information from the past from the neural data structure, and combines it with its previous state and the input to generate the output and the control signal to change the neural data structure content for the next step. The paper presents experimental results on synthetic data showing that the neural data structures provide suitable structured memory to draw the 'attention' of the model to the relevant part of the history. The experiments are performed across a range of transduction tasks, and compared with LSTM as the baseline.

I like the paper and I believe it present interesting neural data structures to implement tractable structured memory networks. However, there are some issues there: - The presentation of the model is not good, i.e. it is very difficult to understand. Authors need to improve the presentation to make it understandable for more people.

- Authors claim the neural Turing machines are intractable, hence justifying the need for more restricted but tractable models of computation. They need to show the intractability of neural Turing machines and the tractability of neural data structures on some simple experiments to back up their claim.

- Authors need to compare in more details with the Facebook work, both in the 'Related Work' section AND in the experiments section. I would very much like to see how these works are compared empirically. - Why the number of baseline LSTM layers is different across tasks and how it has been specified for each task? - Finally, authors need to show the usefulness of the neural data structures on real life datasets, even small. For example, they could have used BTEC for machine translation, if they cannot run the models on large bilingual corpora.

I would very much appreciate authors' comments on the above issues.
Summary: This is a good paper which presents interesting (arguably novel) neural data structures. The only issue is the experiments which are only on synthetic data and not on real datasets.

Author Feedback
Author rebuttal: We thank the reviewers for their excellent feedback.

Reviewer 1:
The alternative Stack/Queue/DeQue formulation suggested by Reviewer 1, using sampling over nondeterministic operations on discrete structures, is definitely worth investigating and comparing to in future expansions to this work, and we will do so. The reason for focussing on end-to-end differentiable structures in this work, which we show solve the tasks presented here, was that they do not encounter some of the high-variance problems which approaches involving sampling over discrete action spaces typically faced. As such, this approach seemed more appealing as a starting point for investigating this class of models, but we do not wish to claim that this is the only, or even the best, way of using external memory. Thanks for the great suggestion for future work.

Reviewer 3:
NTMs and other random access neural memories are more powerful and expressive than the neural Stack/Queue/DeQue memories presented here. We focus on exploring the middle ground between LSTMs and random access networks like NTMs, without making claims about the efficiency or tractability of the more powerful models.

The Facebook AI Research work on continuous Stacks was done concurrently with this work, and probably under review at the same conference. We plan direct comparison in future work, although it was not possible at present.

We will work on readability significantly when preparing the camera ready, notably by giving full sequence examples of the stack operation on a reversed/copied sequence in the appendix. We hope this will make the model presentation easier to follow.

Regarding LSTM benchmark hyperparameters, we grid-searched across a number of layer size and depth options to be maximally fair with regard to the LSTM benchmarks. The best models were selected based on training perplexity, and compared. In practice, this provides a reasonable estimate on the upper bound of LSTM benchmark performance.

Reviewer 4:
Memory-enhanced RNNs and attention mechanisms are complementary ways of attacking the information bottleneck present in standard sequence-to-sequence models, and can be used in unison. We will make note of this in the paper, and future work will certainly investigate how these two approaches can cooperate. Thanks for suggesting that we point this out, as it is very relevant.

Reviewer 6:
We agree that it would be helpful for us to offer more detail about the synthetic data generation process. We will release scripts to reproduce the data, and add more detail in the appendix, as we think these experiments will provide useful "unit tests" for the developers of similar models.

All Reviewers:
All reviewers comment that comparison on "real" data would have been beneficial. We agree, and assure them that evaluation on transduction tasks such as neural machine translation is in the works. However, we wish to point out again that the scale of the synthetic datasets used here is huge: 10e135 training sequences per task. The data will never be traversed more than once, if at all, preventing overfitting. Some of the transduction tasks furthermore exhibit little to no local regularities in the data, requiring models to fully exploit long range dependencies to solve them. Therefore while some aspects of these tasks are easier than natural language data, others are harder: for instance, a target sequence language model cannot bear the brunt of the generative process in bigram flipping, reversal, etc, as there are no local regularities to exploit. Thus the use of synthetic data allows us to explore, compare, and contrast the capabilities of LSTMs with and without external memory on specific aspects of transduction. That said, we do not dispute that the real litmus test of such models is performance on end-to-end natural language processing tasks, and assure reviewers this is planned for further work.

We again thank the reviewers for their time and insightful comments. We feel many suggestions have been made which not only will help us ameliorate the current presentation of the model and experiments and link the present work to other related work, but also provide excellent directions to investigate in follow-up work.